# MEMS Actuators for Optical Microendoscopy

**DOI:** 10.3390/mi10020085

**Published:** 2019-01-24

**Authors:** Zhen Qiu, Wibool Piyawattanametha

**Affiliations:** 1Department of Biomedical Engineering, Institute for Quantitative Health Science and Engineering, Michigan State University, East Lansing, MI 48824, USA; qiuzhen@egr.msu.edu; 2Departments of Biomedical Engineering, Faculty of Engineering King Mongkut’s Institute of Technology Ladkrabang (KMITL), Bangkok 10520, Thailand

**Keywords:** MEMS actuators, microendoscopy, confocal, two-photon, wide-filed imaging, photoacoustic, fluorescence, scanner

## Abstract

Growing demands for affordable, portable, and reliable optical microendoscopic imaging devices are attracting research institutes and industries to find new manufacturing methods. However, the integration of microscopic components into these subsystems is one of today’s challenges in manufacturing and packaging. Together with this kind of miniaturization more and more functional parts have to be accommodated in ever smaller spaces. Therefore, solving this challenge with the use of microelectromechanical systems (MEMS) fabrication technology has opened the promising opportunities in enabling a wide variety of novel optical microendoscopy to be miniaturized. MEMS fabrication technology enables abilities to apply batch fabrication methods with high-precision and to include a wide variety of optical functionalities to the optical components. As a result, MEMS technology has enabled greater accessibility to advance optical microendoscopy technology to provide high-resolution and high-performance imaging matching with traditional table-top microscopy. In this review the latest advancements of MEMS actuators for optical microendoscopy will be discussed in detail.

## 1. Introduction

Actuation and scanning mechanisms have played important roles in novel microendoscopic imaging systems. Common challenges in the development of these miniature instruments are in both design freedom and the integration of miniaturized opto-mechanical components. Microelectromechanical systems (MEMS) fabrication technologies play a valuable and instrumental role in solving the aforementioned issues in order to achieve similar performance as traditional microscopy counterparts. In optical microendoscopy, actuation and scanning mechanisms enable three-dimensional (3D) image formation in the tiny devices with ultra-compact form factors. The technical challenges in designing such kinds of components include the generation of a distortion-free scanning pattern with sufficient speed to mitigate in vivo motion artifacts with millimeter package dimensions. To realize in vivo tissue imaging on living subjects, 5 Hz or faster frame rates are usually required to accommodate movements induced by several factors, such as respiratory displacement, heart beating, and organ peristalsis.

The size of the MEMS actuators determines their mounted locations at either the proximal or distal end of the instrument. A much greater control of the focal volume, including axial scanning for imaging into the tissue, can be achieved with the MEMS actuators positioned distally. However, their typical sizes mounted at the distal end of the instrument should be less than 5.5 mm (for example, fitting the Olympus therapeutic endoscope) in order to be compatible with the tool channel of a standard medical endoscope [1].

A spiral scanner consists of a tubular piezoelectric actuator (small diameter <1 mm) that drives the distal tip of a single-mode (SM) optical fiber using modulated sinusoidal waveforms around the resonant frequency [2]. This approach has been successfully implemented in the scanning fiber endoscope (SFE) with wide-field fluorescence imaging and in representative multi-photon microendoscopes. Compared to the raster scanning, the spiral scan pattern can achieve higher frame rates with large amplitude. Micro-motor-based rotational scanning mechanisms steer the laser beam circumferentially around the longitudinal axis of the endoscope after a 45° deflection off a reflective mirror or prism. Similar methods have been widely used in optical coherence tomography (OCT) and photoacoustic (PA) endoscopes. The galvo-scanner, reliable but aging technology, is a meso-scale electromechanical mechanism that performs beam scanning by deflecting a mounted mirror coated with aluminum or gold depending on the effective wavelength range. The relatively large size limits its use to steering a focused beam into the proximal end of a coherent bundle of optical fibers. This technique has been used in fiber bundle-based confocal microendoscopy (Cellvizio, Mauna Kea Technologies, Paris, France) [3]. The bulky galvo scanner at the proximal end provides deflections in the slow axis and is used with a resonant mirror that performs fast-axis scanning.

Micromirrors have been developed with microelectromechanical systems (MEMS) technologies that use either electrothermal or electrostatic actuators to achieve large deflection angles and high dynamic bandwidths with excellent linearity. These scanners can be batch fabricated on silicon wafers to achieve devices with relatively high yield. MEMS actuators and scanners usually require sophisticated micro-fabrication processes but have great flexibility in scanning speed and device dimensions/geometries. Based on different working principles, there are some typical types of actuation mechanisms for MEMS actuators in microendoscopes [4,5,6,7], including bulky piezoelectric tubing, electrostatic comb-drive or plates, electro-thermal, electro-magnetic, and thin-film piezoelectric materials. Thus far, electrostatic devices are most commonly applied in microendoscope development because of their relative fast scan speeds with ultra-low power consumption (from very low current consumption). However, one of the disadvantages of the electrostatic MEMS devices, relative high driving voltage (>100 V), might become a major concern for clinically translational use in the event of electrical short circuit from the MEMS devices. Recently, some novel electrostatic resonant 2D scanners operating at low driving voltage can potentially resolve this issue. On the other hand, the novel thin-film piezoelectric materials actuated MEMS devices have excellent performance with relatively low driving voltage (<20 V) and ultra-low power consumption and may be the future for scanners and actuators in endomicroscopy [8,9,10]. The low fill-in factor (<50%) of existing micromirrors is a common problem that may be solved by micro-assembly or advanced manufacturing processes [7].

This review is intended to present some representative examples of many exciting optical microendoscopy being pursued around the world based on MEMS actuators.

## 2. Overview of Optical Imaging Modalities

Various optical imaging modalities implemented so far with MEMS actuation technologies will be given as a preview here, including OCT, PA, confocal microscopy (CM), and multiphoton microscope.

By using the short coherence length of a broadband light source, the resolution of OCT can achieve around 1 μm to 15 μm, which usually depends on the light source employed, including supercontinuum white light lasers. OCT imaging system usually has deep tissue penetration up to 3 mm, which is sufficient for imaging of the whole epithelial layer. Furthermore, to realize 3D imaging, 2D lateral scans need to be realized by a moving stage or a scanning mirror (SM). Two-dimensional (2D) MEMS actuators are often used for 3D imaging, although there are some alternative approaches with conventional cable-based actuation and scanning mechanisms driven by rotation motors and pulling stations. The OCT imaging system has been successfully applied to ophthalmology and widely used in clinics since its first seminal report [11]. In addition, researchers have made tremendous efforts on the miniaturization and clinical translation of the OCT system for many kinds of applications on hollow organs [12], including colon, esophagus, bladders, and lung.

Recently, photoacoustic (PA) microscopy has become an emerging and promising imaging technology with several advantages [13], in terms of the tissue penetration, high spatial resolution, and contrast. Based on the light-tissue-sound interaction, the novel imaging modality acquires 3D images through detecting an acoustic wave generated by rapid thermoelastic expansion induced by pulsed laser beam absorption in the tissue specimens. In summary, two representative photoacoustic microscopic imaging systems have been demonstrated, including optical resolution photoacoustic microscopy (so called OR-PAM) and acoustic resolution photoacoustic microscopy (so called AR-PAM). MEMS actuation technologies enable compact photoacoustic microendoscopes and have been developed by several research groups, which will be introduced in detail later.

Based on the continuous-wave (CW) laser, confocal microscopy provides imaging contrast with subcellular resolution optical sectioning capability via a pinhole [14]. The confocal microscope acquires 3D images by stacking two-dimensional lateral (x-y) images (plane by plane) along the z-axis, by utilizing a two-dimensional galvo-scanner and piezo-actuated objective lens. Due to the conventional pre-objective scanning configuration, only a limited field-of-view (FOV) can be realized with a high numerical aperture (NA). Reflective and fluorescence imaging modes are often used for confocal microscopy. While operating in the reflective mode, the confocal microscopy relies on the backscattered light from the tissue and provide morphological information of tissues. On the other hand, fluorescence imaging in conjunction of fluorophore labeling will provide a signal from microstructures with relatively high specificity, high sensitivity, and bright contrast. Bench-top commercial confocal microscopic imaging system with bulky objective lenses have been widely used for ex vivo or intra-vital imaging in biomedical research laboratories. By taking advantage of the MEMS-based actuation technologies, confocal microendoscopes have been developed based on different architectures, including conventional single-axis with pre-objective scanning and new dual-axis with post-objective scanning [1], which will be introduced later in the section on MEMS-based confocal microendoscopy. The traditional single-axis architecture uses high-NA objective lenses that provide high resolution, but limit the working distance significantly. In the novel dual-axis confocal architecture, relatively larger FOV and longer working distance can be realized by the system design consisting of a low numerical aperture lens and scanning elements located right at the post-objective position.

Different from the working principle of confocal microscopy, the two-photon microscopic imaging system is mainly based in nonlinear light-tissue interactions [15,16]. The fluorescence emission signal can be generated while two lower-energy (longer wavelength) photons in the near-infrared (NIR, more than 800 nm) regime arrive at tissue biomolecules simultaneously. To collect the weak fluorescence emission signal, sensitive detectors, such as avalanche photodiodes (APDs) and photomultiplier tubes (PMTs), are usually employed. High peak intensity femtosecond (fs) pulse lasers (~10–200 fs) are required for the two-photon microscopic imaging because the probability of the simultaneous two photon absorption by a single fluorophore is relatively low. With a high numerical aperture objective lens, the fluorescence emission signal will be acquired only from the focus plane. In addition, with the longer wavelength used in two-photon microscopic imaging, it enables deep tissue penetration and stronger imaging contrast with relatively lower photo-bleaching and photo-damage to the tissue specimens. Thanks to the rapid development of advanced fiber optics and fiber laser technologies, ultra-compact MEMS-actuated two photon microendoscopes have been successfully demonstrated for basic biological studies (like neural circuit imaging) and clinically translational applications [17]. For delivering a high-intensity, ultrafast pulse laser with minimal distortion, photonic crystal fibers (PCFs) have been developed and widely used in microendoscopic imaging systems [17].

## 3. MEMS-Based Optical Coherence Tomography (OCT) Microendoscopy

Since the first electro-thermal MEMS 2D scanner-based OCT endoscopic prototype [18] was demonstrated, researchers have developed several types of MEMS-based OCT microendoscopes [19,20]. Although traditional electromagnetic micro-motor based OCT catheters have already been widely used for in vivo imaging on small animal models or human patients [21,22,23], mass-producible MEMS actuator-enabled OCT microendoscopes will very likely become the future trend because MEMS-based micro-devices have many advantages, especially in terms of the miniaturization potential and repeatability. MEMS-based micro-devices may be based on various working principles and actuation mechanisms, such as electrostatic [24,25,26,27,28,29], electrothermal [18,30,31,32,33], bulky PZT-based fiber scan tube [34,35,36,37,38,39], and electromagnetics [40].

Among the variety of micro-devices, the electrostatic comb-drive-actuated MEMS scanner is a popular one [24,25,26,27,28,29]. For example, one of the representative MEMS-based OCT microendoscopes, as shown in Figure 1 [28], has utilized an electrostatic MEMS scanner [25] driven by angled vertical comb (AVC) actuators for a large tilting angle. The effective mirror aperture’s diameter is as large as 1 mm, which is sufficient to reflect the light beam for side-view high-resolution imaging with the MEMS-based endoscopic catheter. The single-mode fiber (SMF), fiber collimator, and the MEMS scanner are fully integrated in an aluminum-based packaging. The detailed fiber-based optical system design of the time-domain OCT imaging system and the real-time data acquisition system with high sampling speed are illustrated in Figure 2a. Three-dimensional OCT image volumes acquired in vitro from a hamster cheek pouch are shown in Figure 2b. Both horizontal (also called *en-face*) and vertical cross-sectional plane images extracted from the 3D OCT volume, in Figure 2c, have demonstrated high-resolution morphological changes inside the tissue specimen.

Most recently, a novel MEMS-based OCT microendoscope with circumferential-scanning has been developed by the engineering team led by Xie from the University of Florida [41] through a unique optical design utilizing multiple electrothermal MEMS scanners. An array of ultra-compact electro-thermally actuated MEMS scanners (Figure 3a) are integrated at the distal end of the catheter to reflect collimated beams, as shown in Figure 3b,c. Flexible printed circuit boards (FPCB) provide driving current for electrothermal scanners. All of the micro-optical components and MEMS-based circumferential scanning systems have been fully integrated and assembled in a compact form factor (Figure 3e) for potential in vivo imaging application in the human gastrointestinal (GI) tract. The fiber-based collimating system is used for laser excitation, as shown in Figure 3f.

## 4. MEMS-Based Photoacoustic Microendoscopy

A MEMS scanner-based photoacoustic microscope (PAM) system’s conceptual design has been demonstrated by Chen [42] by taking advantages of both an optical micro-ring resonator and electrostatic comb-drive-actuated MEMS scanner. The ultrasensitive micro-ring resonator with broad bandwidth, developed by Ling [43], is one type of micro-/nano-photonic device which sense an ultrasonic signal using optical approaches. As shown in Figure 4a, a fiber-based optical system setup with pulse laser excitation (wavelength 532 nm), MEMS mirror driving system, real-time data acquisition system has been described in the schematic drawing. The electrostatic MEMS scanner within the package, in Figure 4b, provides the lateral laser beam point-scanning in raster scanning mode at a slow rate. This new PAM imaging system can provide ex vivo optical resolution photoacoustic images of the tissue. To detect the weak photoacoustic signal, the micro-ring resonator is located right under the tissue specimen with acoustic signal coupling media, such as water or ultrasonic gel.

To realize the photoacoustic microscopic system in more portable or endoscope-friendly form factor, researchers have been making tremendous efforts on the miniaturization of the imaging system design and the distal scanhead with MEMS technologies. A new handheld photoacoustic microscope (PAM) probe [40], as shown in Figure 5, has been developed recently for potential clinical application. The distal scanhead of the handheld PAM system, 17 mm in diameter and a weight of 162 g, mainly consists of the fiber-based collimator, ultrasound detector, acoustic and photonic beam coupler, and beam scanning system. The handheld PAM system has integrated a newly custom-developed electromagnetic MEMS 2D scanner, shown in Figure 5a. The schematic drawing of the full imaging system is illustrated in Figure 5b, including the high-speed data acquisition system, ultrasonic transducer, and fiber based optics. High-resolution imaging quality with a large FOV using this handheld PAM system has been demonstrated by imaging the blood vessel of a mouse ear, shown in Figure 6. The PAM imaging system has also been used to delineate a human mole to demonstrate its clinical application in delineating melanoma which has the highest death rate among skin cancers and may cause about 9730 deaths the United States.

Based on an advanced ultra-compact electrothermal MEMS 2D scanner, a novel miniaturized MEMS-based photoacoustic (PA) microendoscope has been recently developed through collaborative team work led by Xi and Xie [44]. This work has been the most advanced PA microendoscopic imaging system and is close to clinical application. The cross-sectional view photograph of the endoscopic packaging is shown in Figure 7a, including a GRIN lens-based fiber-based collimator, electrothermal MEMS scanner, and optical and acoustic coupler. The new MEMS-based PA microendoscope can acquire high-resolution photoacoustic images of tissue specimens. The image performance of this new photoacoustic microendoscope has been demonstrated on a mouse ear, as shown in Figure 8c.

Thus far, several miniaturized photoacoustic microendoscopes have been developed using electromagnetic [46,47,48] or electrothermal scanners [44]. To co-axially steer both laser and ultrasonic beams, a water-immersible electromagnetic MEMS scanner [48] has been custom-made to operate in the ultrasound coupling media. Due to the high detection sensitivity and broader bandwidth with very compact form factor, a micro-ring resonator [43,49,50,51,52] has attracted more attention and been fully explored, such as the transparent micro-ring for microendoscope applications [51,52].

## 5. MEMS-Based Confocal Microendoscopy

Compared to other relatively new imaging modalities, confocal microscopy has been studied for decades. MEMS-based confocal microendoscopes were invented a long time ago since the first seminal work demonstrated by Kino and Dickensheets [53]. Later, researchers have focused on improving the lateral or axial resolution and depth imaging while miniaturizing the confocal microendoscopes using MEMS technologies. For instance, the new MEMS-based 3D confocal microendoscope with a tunable Z-focus has been developed by Xie’s team [45] using an advanced electrothermal MEMS scanner with tunable objective lens mounted in the center of the moving Z-axis stage with large translational movement (>300 µm) at low voltage. As shown in Figure 8, the MEMS-actuated tunable objective lens is located at the distal end, which is very close to the tissue specimen. By being fully packaged in stainless steel tubing, the fiber-based microendoscope could potentially be applied for clinical applications on humans (Figure 8b).

Another interesting electrothermally-actuated MEMS fiber scanner has also been invented and fully integrated into the MEMS-based confocal microendoscope [54]. A SEM image of the compact electrothermal MEMS fiber scanner is shown in Figure 9b. As shown in Figure 9, the team from KAIST [54] has recently developed a novel scanning fiber-enabled ultra-thin confocal microendoscope which can be easily inserted into the miniature tool channel of the medical laparoscope, shown in Figure 9c.

Compared to the conventional single-axis confocal architecture [45], the novel dual-axis confocal (DAC) configuration offers superior dynamic range in the Z-axis with higher axial resolution. Based on the fully-scalable DAC optics architecture, miniaturization using 2D/3D MEMS scanners and micro-optics have been performed during the past ten years. Both electrostatic MEMS scanners and thin-film piezo-electrical (PZT: lead zirconate titanate)-based MEMS scanners have been developed for MEMS-based DAC microendoscopy.

A monolithic thin-film piezo-electrical MEMS scanner [55] (footprint in less than 3.2 mm by 3.0 mm), with both vertical (Z-axis) and lateral (X-axis or Y-axis) scanning capabilities, has been demonstrated for the first time to perform horizontal and vertical cross-sectional imaging. The schematic drawing (Figure 10a) illustrates the integration of the thin-film PZT-based MEMS scanner with multidimensional freedom inside the optical design of the DAC microendoscope. A photograph of the thin-film PZT based MEMS device, which provides large translational motion for Z-axis focus change (>200 µm) and wide tilting angle (> ± 5° mechanically) for lateral scanning, is shown in Figure 10b.

Although new thin-film PZT-based MEMS scanners show promising technical advantages over conventional MEMS devices, their micro-machining processes are still challenging due to the complexity of preparation and patterning of thin-film piezo-electrical materials. On the other hand, the traditional electrostatic MEMS scanner [56,57,58,59,60] has recently been fully explored with unique mechanical flexure designs to meet the requirements from 3D confocal microendoscopic imaging systems [61,62,63,64,65]. As shown in Figure 11, a novel monolithic electrostatic MEMS scanner with switchable lateral and vertical scanning capabilities have been successfully demonstrated with a compact footprint (<3.2 mm × 3.0 mm) for DAC microendoscopes. The new electrostatic scanner is based on the parametric resonance working principle with an in-plane comb-drive configuration. Through design optimization, the driving voltage can be close to 40 V, which is safe for human patients. With cross-sectional depth imaging, MEMS-based DAC microendoscopes may potentially be used for molecular contrast agent-based multi-color fluorescence imaging [66,67,68,69] for colorectal cancer early detection in the human gastrointestinal tract.

By combining two separate electrostatic MEMS scanners, lateral (XY) and vertical (Z-axis) scanners [70], respectively, a new 3D MEMS scan engine-based DAC microendoscope with multi-color achromatic optics design could perform real-time 3D volumetric imaging in the tissue specimen for both clinical applications and system biology studies on live rodents. Furthermore, monolithic multiple degree-of-freedom or a 3D thin-film PZT-actuated micro-stage [8,9,10] will also potentially provide the 3D imaging without increasing the optical design complexity. As alternative approaches for miniature confocal system design, tunable optics-based [71] and micro-grating-based spectral encoded confocal microendoscopes [72] can realize depth imaging and *en-face* imaging with fewer scanning components.

## 6. MEMS-Based Multiphoton Microendoscopy

Not only being used in OCT and confocal system, electrostatic MEMS scanners have already demonstrated their critical roles for miniaturized multiphoton microendoscopic imaging system development since the first prototype was demonstrated by Piyawattanametha in 2006 for mice brain in vivo imaging [73,74]. Extended applications [75,76,77] have also been studied using the electrostatic MEMS 2D scanner-enabled multiphoton microendoscope, including femtosecond laser-based microsurgery [77].

Recently, handheld and endoscopic multiphoton microscopes have been developed with custom-made electrostatic MEMS 2D scanners [78] and Er-doped fiber laser [79,80]. For example, as shown in Figure 12, a new MEMS based two-photon fluorescent microendoscope [78] with a compact distal end is packaged in the stainless steel tube. The 2D MEMS resonant gimbal-based scanner can perform a lateral scan around the X- and Y-axes. With administration of Hoechst (nucleic acid stain), in vivo fluorescence imaging has been demonstrated in the distal colon of CDX2P-NLS Cre;adenomatosis polyposis coli (CPC;Apc) mouse model, which mimics human colorectal cancer diseases, as shown in Figure 13. A single-frame from a video sequence is shown in Figure 13a while the post-processed image after averaging (5 frames) is shown in Figure 13b. Compared to the images of H and E slides, the sub-cellular high-resolution microscopic imaging system could potentially provide histology-like imaging.

Due to the footprint size of the electrostatic MEMS scanner, the distal end of the microendoscopic scanhead could not be easily miniaturized to less than 2.0 mm. However, the bulk PZT tube-based fiber scanner could potentially be fabricated with an ultra-thin wall and an outer diameter less than 1.5 mm so that the piezo tube fiber scanner enabled multiphoton microscope’s distal end could be very small. Fiber scanner-based miniaturized multiphoton microscope was first demonstrated by Helmchen and Denk in 2001 for in vivo imaging on rodents’ brains [81]. In addition, similar to the very small piezo tube-based fiber scanner [82,83], a bulk piezo sheet-based 2D raster-mode fiber scanner has also been investigated for multiphoton microendoscopic label-free imaging on unstained tissue specimens [84].

Based on the extensive experience on the multiphoton imaging system development, the team led by Li at Johns Hopkins University has recently developed a novel piezo tube-based fiber scanner-enabled miniaturized two-photon and second harmonic imaging system [85]. Aimed for label-free functional histology in vivo, the new fiber-optic scanning two-photon endomicroscope mainly consists of several key components, including a flexible double cladding fiber (DCF) for laser excitation and harvesting emission light, GRIN lens, and the very small piezo tube-based fiber scanner. A miniaturized custom-made objective with longitudinal focal shift has been developed by collaborating with GRINTECH (GmbH, Jena, Germany). A phase diffractive grating is sandwiched between two GRIN elements, as shown in Figure 14b.

The novel fiber optic multiphoton microendoscope developed by Liang [85] performs both two-photon fluorescence (2PF) and second harmonic generation (SHG) label-free structural imaging in vivo on small animal and human patients. As shown in Figure 15a,b, the overlay of intrinsic 2PF and SHG images have been acquired ex vivo from mouse liver. The emission signal was detected through two spectral channels: 496–665 nm (green, 2PF signal) and 435–455 nm (red, SHG signal). Figure 15c,d show the in vivo two-photon auto-fluorescence images of the mucosa of mouse small intestine, while the two detection channels are 417–477 nm for NADH (green) and 496–665 nm for FAD (red). Time-lapse SHG images of a cervical collagen fiber network have been acquired through intact ectocervical epithelium of cervices dissected from preterm-birth mouse models Figure 15e and normal pregnant mice (Figure 15f) at gestation day 15.

Electrostatic comb-drive actuated MEMS scanner and piezo fiber scanners have demonstrated their great potential in the development of multiphoton microendoscopes. In addition to those two actuation mechanisms, electrothermal MEMS scanners [86,87] have also been proposed for fiber scanning in the multiphoton imaging system although it may not be quite ready for clinical applications yet.

## 7. Fluorescence Wide-Field Endoscopy

Since its first application in the scanning-probe microscopes, miniaturized piezo tube based fiber scanner have demonstrated its great potential in several optical imaging modalities, such as OCT [34,36], or multiphoton [82,88]. By taking advantages of its ultra-thin form factor, the piezo tube has also been used in the scanning fiber endoscope (SFE) for wide-field imaging with both reflective [2] and fluorescent modes [89]. Compared to other MEMS scanner-based fluorescence imaging systems [90], the SFE-based endoscopic imaging system could have a much smaller outer diameter. Recently, Savastano and Zhou have demonstrated multimodal laser-based angioscopy [91] for structural, chemical, and biological imaging of atherosclerosis using the miniature catheter imaging system. As shown in Figure 16a, the scanning fiber endoscope excites tissues by scanning blue (424 nm), green (488 nm), and red lasers (642 nm) in a spiral pattern. Backscattered (reflectance) light and the fluorescent signal is collected by a ring of multimode fibers located in the periphery of the scanner housing and shaft and conducted to a data acquisition computer for image reconstruction. The optical system can be packaged with an outer diameter of 2.1 mm (left) or 1.2 mm (right) endoscopes, as shown in Figure 16b.

A description and summary of performance for several different MEMS scanning mechanisms that are currently being developed for in vivo endomicroscopy are summarized in Table 1.

## 8. Conclusions

In this review, we present a review of the latest advancements of MEMS actuator-based optical microendoscopy. High precision in manufacturing coupled with various optical/mechanical functionalities derived from MEMS fabrication techniques make these components well suited to integrate into many optical based microendoscopy. Nonetheless, it is crucial to gain understandings of other underlying principles, such as life-cycle, sizes, speed, material properties, force, operating ranges, and power consumptions to achieve optimum performance before uniting all components altogether. In addition, clinical considerations, such as usage simplicity and ergonomics, cannot be overlooked as those parameters are used to dictate the overall designs and selections of MEMS actuators for optical microendoscopy. Overall, electrostatic-based actuation is one of the most popular actuators employed in endoscopic-based imaging despite the fact that it provides both relatively moderate actuation force and limited scanning ranges. However, the advantages are their ease of fabrication, lower complexity of integration to endoscope packages, and relative fast scanning speed to help reduce motion artifacts. All in all, MEMS actuator-based optical microendoscopy has been showing great promise to deliver high-performance imaging on par with traditional microscopy in aiding medical diagnosis procedures.

## Figures and Tables

**Figure 1 micromachines-10-00085-f001:**
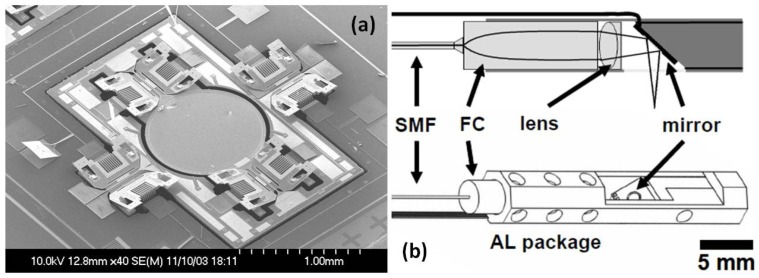
(**a**) Scanning electron micrograph of the MEMS two-axis optical scanner. The scanner has a large 1 mm diameter mirror and uses angled vertical comb (AVC) actuators to produce a large angle scan for high-resolution imaging. (**b**) Optical schematic and mechanical drawing of the optical coherence tomography (OCT) catheter endoscope. SMF: single mode fiber, FC: fiber collimator, AL: aluminum. Reproduced with permission from [28]; published by OSA, 2007.

**Figure 2 micromachines-10-00085-f002:**
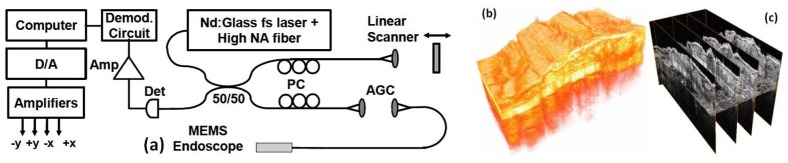
(**a**) OCT system schematic for imaging with the MEMS endoscope. PC: polarization control, AGC: air-gap coupling, Det: detector, Amp: amplifier, D/A: digital to analog Converter; OCT images acquired with the MEMS scanning catheter. (**b**) Three-dimensional rendering of the OCT volume dataset from a hamster cheek pouch; and (**c**) the serial cross-sections and *en-face* plane extracted from a 3D OCT volume of a hamster cheek pouch acquired in vitro. Reproduced with permission from [28]; published by OSA, 2007.

**Figure 3 micromachines-10-00085-f003:**
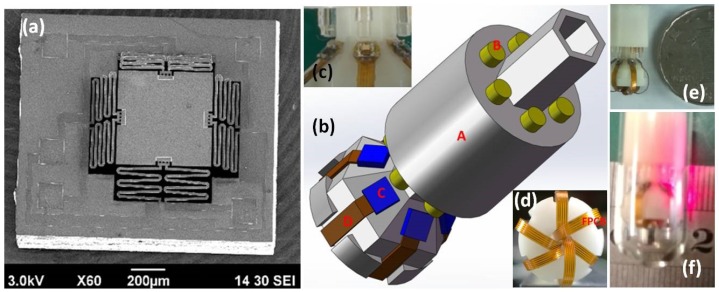
Electrothermal MEMS scanner-based OCT catheter. (**a**) A SEM image of the MEMS scanner. (**b**) The schematic 3D drawing of the probe design. A: 3D-printed probe head; B: C-lens collimators; C: the MEMS chips; D: FPCB. (**c**) A zoom-in picture showing the C-lens collimators and MEMS chips. (**d**) A back-view picture showing the flexible printed circuit boards (FPCBs) folded into the hollow hole. (**e**) A photo of the assembled probe (pictured with a Chinese Yuan coin). (**f**) Photograph of the OCT probe with laser beam scanning. Reproduced with permission from [41]; published by OSA, 2018.

**Figure 4 micromachines-10-00085-f004:**
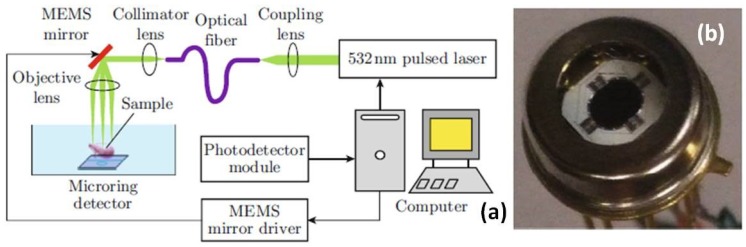
MEMS-based optical resolution photoacoustic microscope. (**a**) Schematic of the MEMS-based optical resolution photoacoustic microscopy (OR-PAM) system and (**b**) photography of the MEMS mirror. Reproduced with permission from [42]; published by OSA, 2012.

**Figure 5 micromachines-10-00085-f005:**
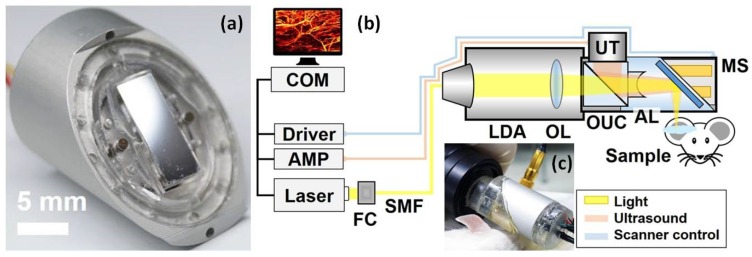
Handheld photoacoustic microscopy (PAM) probe. (**a**) Electromagnetic scanner. (**b**) Schematic of the handheld system. AL: acoustic lens; AMP: amplifier; COM: computer; EM: electromagnets; FC: fiber collimator; LDA: light delivery assembly; MS: MEMS scanner; OL: objective lens; OUC: opto-ultrasound combiner; SMF: single mode fiber; UT: ultrasonic transducer. (**c**) Photograph of the handheld PAM probe. Reproduced with permission from [38]; published by Nature, 2017.

**Figure 6 micromachines-10-00085-f006:**
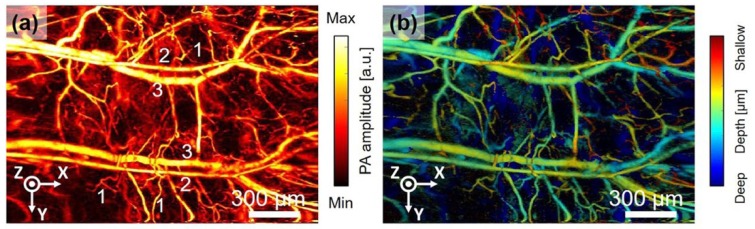
Extension of field-of-view (FOV). (**a**) Single PA MAP image of a mouse ear with an extended FOV (2.8 × 2 mm). (**b**) Depth map image corresponding to PA MAP image (**a**). Reproduced with permission from [38]; published by Nature, 2017.

**Figure 7 micromachines-10-00085-f007:**
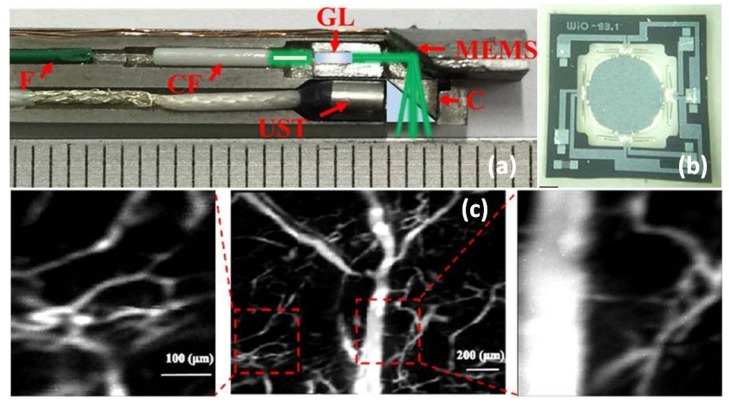
MEMS-based photoacoustic (PA) microendoscope. (**a**) Photograph of the assembled imaging probe. (**b**) Photograph of the electrothermal MEMS mirror. F: fiber; CF: ceramic ferrule; GL: GRIN lens; MEMS mirror; UST, ultrasound transducer; C, cube. (**c**) High-resolution photoacoustic imaging of a mouse ear, the MAP image (left) was acquired with 20 sub images and two typical subimage (right) indicated by the dashed rectangle. Reproduced with permission from [44]; published by OSA, 2017.

**Figure 8 micromachines-10-00085-f008:**
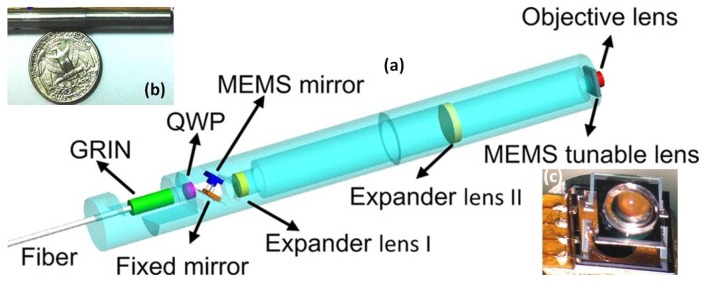
Electrothermal MEMS scanner-based confocal microendoscope. (**a**) Schematic 3D drawing of the microendoscope design. (**b**) Photograph of the assembled microendoscopic probe with US coin as a reference. (**c**) Photograph of an assembled tunable lens on the electrothermal MEMS scanner. Reproduced with permission from [45]; published by Elsevier, 2014.

**Figure 9 micromachines-10-00085-f009:**
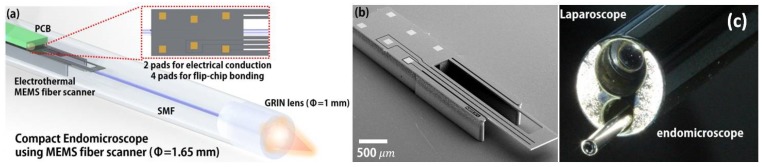
Electrothermal MEMS fiber scanner-based confocal microendoscope. (**a**) Schematic drawing of the catheter with a Lissajous scanner. The MEMS fiber scanner is fabricated for flip-chip bonding, which minimized the electrical packaging dimensions, resulting in the scanner package being 1.65 mm in diameter. (**b**) Top-view SEM image of the fabricated microactuator. The footprint dimension indicates 1 mm × 5 mm × 0.43 mm; (**c**) Photograph of the packaged catheter assembled to the laparoscopic functional channel. Reproduced with permission from [54]; published by OSA, 2018.

**Figure 10 micromachines-10-00085-f010:**
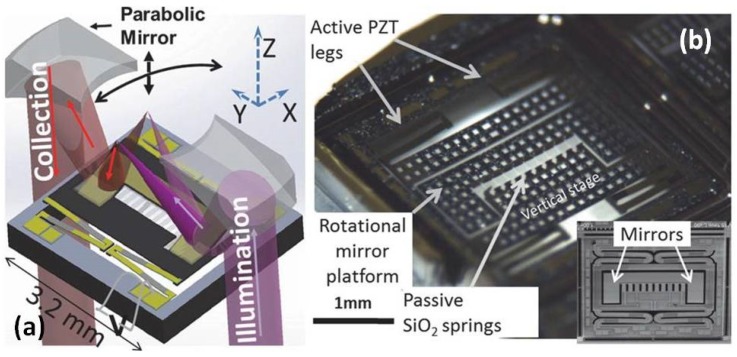
MEMS-based dual-axis confocal (DAC) microendoscope. (**a**) Schematic drawing of the MEMS vertical-rotational stage scanning for cross-sectional imaging with the DAC microendoscope. (**b**) Photograph of the vertical-rotational MEMS scanning stage based on an active outer vertical displacement and a passive inner resonant scanning. Inset: Variant with solid dog-bone mirror surface for the DAC microscope. Reproduced with permission from [55]; published by IEEE, 2014.

**Figure 11 micromachines-10-00085-f011:**
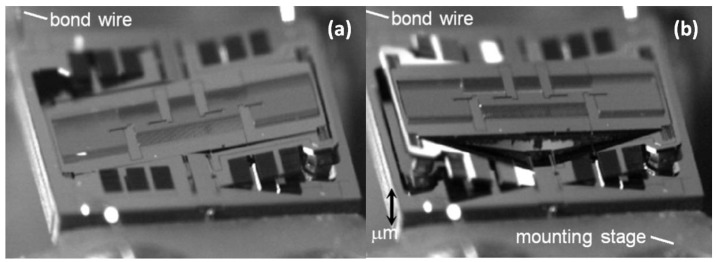
Electrostatic MEMS scanner with switchable lateral and vertical scanning capability. (**a**) Demonstration of the lateral X-axis and Y-axis scanning for horizontal cross-sectional imaging. (**b**) Demonstration of the translational Z-axis scanning for vertical cross-sectional imaging. Reproduced with permission from [48]; published by OSA, 2016.

**Figure 12 micromachines-10-00085-f012:**
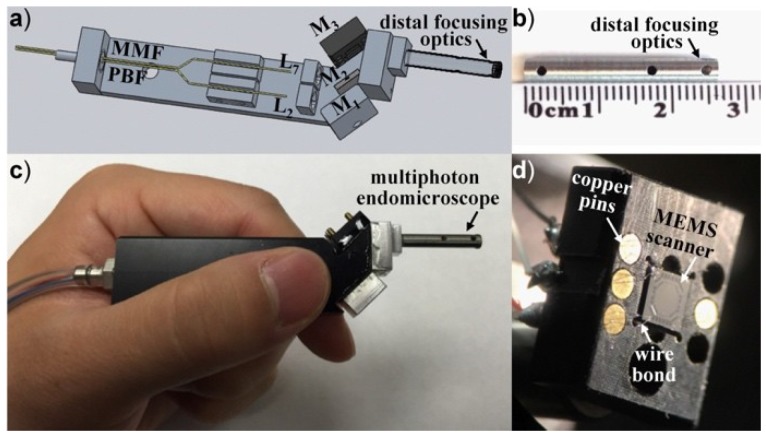
MEMS-based two-photon microendoscope. (**a**) Schematic CAD drawing of the optical circuit. (**b**) Distal end of focusing optics. (**c**) Handheld instrument which is used to perform repetitive imaging in small animal models of human disease. (**d**) Photograph of the MEMS scanner wire bonded on the substrate. Reproduced with permission from [78]; published by OSA, 2015.

**Figure 13 micromachines-10-00085-f013:**
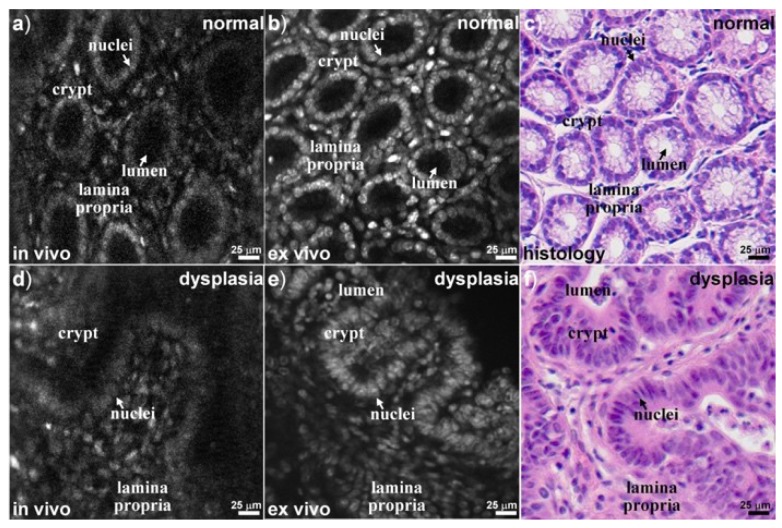
Fluorescent imaging results using the MEMS-based two-photon microendoscope. (**a**) Two-photon excited fluorescence image of normal colonic mucosa. (**b**) In-vivo image from normal averaged over five frames. (**c**) Corresponding histology of normal colon. Single frame from video of dysplastic crypts from colon of CPC; adenomatosis polyposis coli (Apc) mouse (**d**) In vivo at 5 frames/s and (**e**) In vivo image (averaged over five frames). (**f**) Corresponding histology of dysplasia. Reproduced with permission from [78]; published by OSA, 2015.

**Figure 14 micromachines-10-00085-f014:**
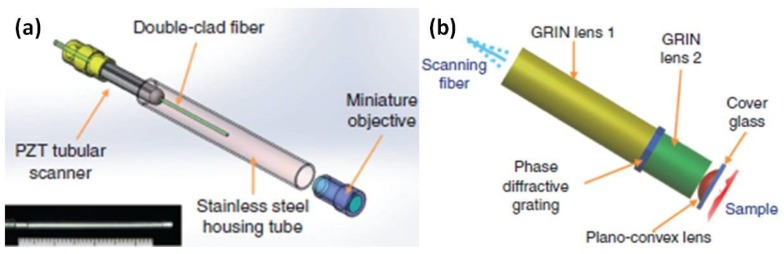
Fiber optic scanning two-photon endomicroscope. (**a**) Schematic drawing of the mechanical assembly based on double cladding fiber (DCF), the bottom inset is the photograph of the endomicroscope with an outer diameter of about 2.1 mm and a rigid length of about 35 mm. (**b**) Miniature custom-made objective and longitudinal focal shift. A phase diffractive grating is sandwiched between two GRIN elements. The fiber- and sample-side WDs are designed to be 200 μm in air and water. Reproduced with permission from [85]; published by Nature, 2017.

**Figure 15 micromachines-10-00085-f015:**
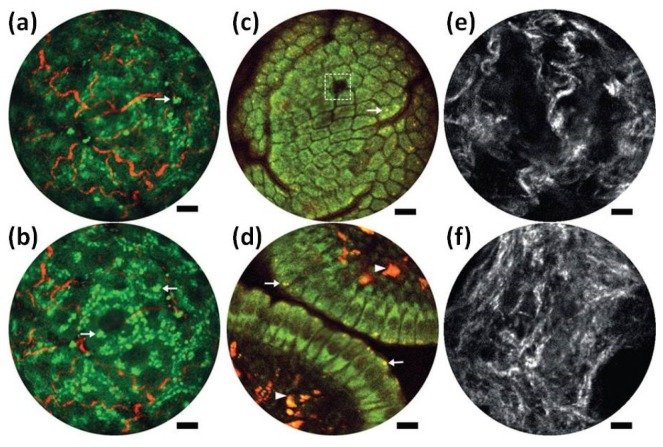
Endomicroscopy two-photon fluorescence (2PF) and second harmonic generation (SHG) label-free structural imaging. (**a**,**b**) Overlay of intrinsic 2PF and SHG ex vivo images of mouse liver. (**c**,**d**) Two-photon autofluorescence in vivo images of the mucosa of mouse small intestine. The two detection channels are 417–477 nm for NADH (green) and 496–665 nm for FAD (red). (**e**,**f**) SHG images of the cervical collagen fiber network acquired through intact ectocervical epithelium of cervices. Scale bars are 10 μm. Reproduced with permission from [85]; published by Nature, 2017.

**Figure 16 micromachines-10-00085-f016:**
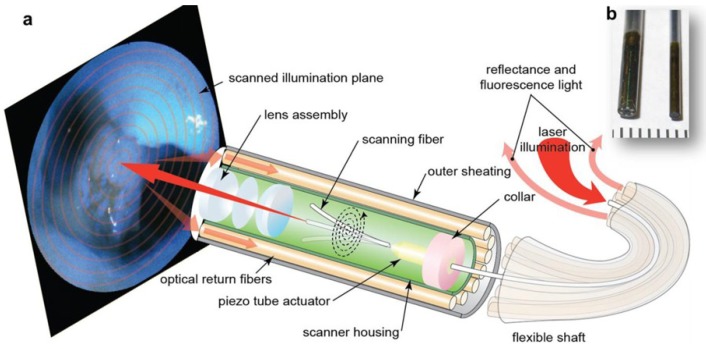
Scanning fiber endoscope optics and fluorescence spectrometry of major structural constituents of atherosclerotic plaques. (**a**) The scanning fiber endoscope excites tissues by scanning blue (424 nm), green (488 nm), and red lasers (642 nm) in a spiral pattern. Backscattered (reflectance) light and fluorescent signal is collected by a ring of optical fibers located in the periphery of the scanner housing and shaft, and conducted to a computer for image generation. (**b**) The optical system can be mounted in 2.1 mm (left) or 1.2 mm (right) endoscopes. Reproduced with permission from [91]; published by Nature, 2017.

**Table 1 micromachines-10-00085-t001:** MEMS scanning mechanisms developed for in vivo optical endomicroscopy.

Imaging Modality	Scan	Res (µm)	FOV	Frame Rate (Hz)	Applications	Advantages	Disadvantages
Fluorescent Wide-Field	Piezo	100–300	~70–90°	~30	GI, respiratory, ear, urinary, reproductive tracts,	High imaging speed, inexpensive laser source, minimal moving parts, commercial devices exist	Relatively low resolution and contrast, no depth sectioning
Single-axis Confocal	Piezo, PZT, Electrostatic, Electrothermal, Magnetic	0.5–5	0–150°	>2	GI, respiratory, ear, urinary, reproductive tracts	High sensitivity provide functional information miniaturization through proximal or distal ends commercial devices exist	Limited contrast and wavelength, limited tissue penetration (<100 µm), limited working distance, increased aberration due to high NA optics
Dual-axis Confocal	Electrostatic	3–6	250–1000 µm	>15	Skin, GI tract, liver, head and neck, pancreas,	Effective out-of-focus rejection of scattered light for high contrast, deep tissue penetration (~400 µm), relatively isotropic resolution	Low NA optics limits sensitivity, challenging alignment of a dual-beam configuration
OCT	Piezo, PZT, Electrostatic, Electrothermal, Magnetic	1–15	2000–5000 µm	>60	GI, respiratory, ear, urinary, reproductive tracts	Impressive miniaturization, high sensitivity, dynamic range, high imaging speed, deep tissue penetration (a few mm)	Label-free imaging, expensive detector array, Short dynamic range along depth
Two-photon	Piezo, PZT, Electrostatic, Electrothermal, Magnetic	0.5–2	200–500 µm	>5	GI, respiratory, tracts	High resolution and contrast, deep tissue penetration (~500 µm ~1 mm) less photobleaching and phototoxicity, Commercial devices exist	Relatively expensive laser source and optics, need dispersion compensation or special fibers to maintain pulse shape
Optical resolution photoacoustic microscope (OR-PAM)	Electrostatic and Electrothermal	~5	1000 µm	10	Breast, brain	High spatial resolution and contrast high imaging speed, deep tissue penetration (a few mm)	Relatively expensive laser source progress on miniaturization is still ongoing

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
