# Peer review of "MEMS Actuators for Optical Microendoscopy"

_micromachines, 2019, doi:10.3390/mi10020085_

Round 1

Reviewer 1 Report

The manuscript “micromachines-406134” entitled " MEMS Actuators for Optical Microendoscopy" submitted for publication in “Micromachines” aims to provide a summary of the most recent advances in the development of MEMS for optical microendoscopy.  This is particularly valuable for the in situ prescreening to detect lesions for next confirmation procedures, promising to save time and costs avoiding the processing of negative cases.  Only few points of the text need attention, as detailed below, to become suitable for publication in “Micromachines”.

In general, the paper is well written. Only the right “position” of abbreviations is to be checked along the text, such as for example MEMS, given in the abstract, used as abbreviation at line 29/30, while as word and abbreviation at line 58. Please check also (OCT), and (PA).

Specific remarks:

- Lines 129-132. The concept of the sentence “With high numerical aperture objective lens, the fluorescence emission signal will be acquired only from the focus plane, which enables deep tissue penetration and stronger imaging contrast with relatively lower photo bleaching and photo‐damage to the tissue specimens” should be improved, because “deep tissue penetration”, “relatively lower photo bleaching and photo‐damage” are ensured by the low energy of the long wavelength used.

- Grammar of sentence at lines 341-342 is to be revised, a verb is missing.

 - Figure 13, legend. The description is to be improved. While in the context of the original paper it can be easily realized (i.e. from the abstract) that two photon fluorescence was taken after stain (Hoechst) administration to mouse, and legend stated clearly that Figure 13 (a) was collected in vivo, such information s missing in the present paper. On the other hand, legend of figure 15 correctly reports that autofluorescence has been measured.

Author Response

Please see the attached pdf file.

Reviewer 2 Report

See the attached

Reviewer 3 Report

It's a nice and comprehensive review paper.
